# The General Trends of Genetic Diversity Change in Alien Plants’ Invasion

**DOI:** 10.3390/plants12142690

**Published:** 2023-07-19

**Authors:** Han Jiang, Yi Zhang, Wenqin Tu, Geng Sun, Ning Wu, Yongmei Zhang

**Affiliations:** 1China-Croatia ‘Belt and Road’ Joint Laboratory on Biodiversity and Ecosystem Services, Chengdu Institute of Biology, Chinese Academy of Sciences, Chengdu 610041, China; jianghan@cib.ac.cn (H.J.); zhangyi921013@163.com (Y.Z.); sungeng@cib.ac.cn (G.S.); 2University of Chinese Academy of Sciences, Beijing 100049, China; 3State Key Laboratory of Earth Surface Processes and Resource Ecology, College of Life Sciences, Beijing Normal University, Beijing 100875, China; wenqintu@163.com

**Keywords:** invasive alien plants, meta-analysis, invasiveness, risk assessment, dynamic prediction

## Abstract

Genetic diversity is associated with invasion dynamics during establishment and expansion stages by affecting the viability and adaptive potential of exotics. There have been many reports on the comparison between the genetic diversity of invasive alien species (IAS) in and out of their native habitats, but the conclusions were usually inconsistent. In this work, a standard meta-analysis of the genetic diversity of 19 invasive plants based on 26 previous studies was carried out to investigate the general trend for the change of IASs’ genetic diversity during their invasion process and its real correlation with the invasion fate. Those 26 studies were screened from a total of 3557 peer-reviewed publications from the ISI Web of Science database during the period of January 2000 to May 2022. Based on the selected studies in this work, a general reduction of IASs’ genetic diversity was found in non-native populations compared to that in native ones, while the difference was not significant. This finding suggested that regardless of the change in genetic diversity, it had no substantial effect on the outcome of the invasion process. Therefore, genetic diversity might not serve as a reliable indicator for risk assessment and prediction of invasion dynamic prediction in the case of IASs.

## 1. Introduction

Biological invasion is a process in which alien species invade a new region from its native habitat naturally or artificially, causing great ecological degradation and economic losses [1]. The beginning of biological invasions could date back to the discovery of America by Columbus in 1492 [2], and it has become one of the most serious environmental events worldwide [3]. The total reported monetary expenditures for invasion management reached a minimum of USD 1.288 trillion over the past five decades (1970–2017) [4]. The study of the mechanism of invasion is of crucial significance for its management and, thus, to avoid the negative influences caused by invasive alien species (IAS).

The process of a successful invasion of an IAS can be divided into three steps (introduction, establishment, and expansion) [5], all of which are dependent on the IAS’s own biological characteristics, including wide environmental tolerance, high phenotypic plasticity level, and self-reproduction ability, effective dispersal capability, high relative growth rate, high competitiveness and/or avoidance of genetic bottlenecks following founder effects [6]. All these characteristics representing the invasive capacity of an IAS are determined by its genes. Genetic analysis is useful to predict whether a non-indigenous species can pass through the stages of establishment and expansion to become invasive species [7]. Among the factors in genetic analysis, genetic diversity, which affects the viability and adaptive potential of the exotics, was considered to be a key determinant of invasion dynamics during those two stages [8,9]. Genetic diversity (also known as genetic polymorphism) is the variation in genetic information between distinct individuals (or chromosomes) of a given species (or population) [10]. It consists of the differences between individuals at different levels, such as morphological features, structure and chromosomal number, and polymorphisms of sequences of DNAs or proteins. Most estimations about genetic diversity have been made based on allele or genotypic frequency obtained by studying molecular markers of a population [11,12]. Increasing evidence shows that genetic diversity provides raw materials necessary for adaptive evolution [13]. Maintaining genetic diversity within natural populations can maximize their potential to withstand and adapt to biotic and abiotic changes. According to this, some researchers considered that increased genetic diversity was beneficial to non-native populations’ biological invasions [14,15]. But other studies indicated declination in genetic diversity did not terminate the invasion; thus, high genetic diversity was not essential for successful introduction [9,16,17]. Hence, conducting a comparative analysis of the genetic diversity and structures of invasive plants in both their native and exotic regions is instrumental in uncovering early demographic establishment and evolutionary adaptation subsequent to invasion [18,19]. This endeavor holds significant implications for the development of more effective preventive and control strategies [20].

Further, there have been plenty of reports on the change in genetic diversity of the alien species at the DNA level after their invasion [20,21,22]; however, quite a lot of inconsistency occurred. Higher genetic diversity was detected in native populations than in the introduced ones in some studies [23,24,25], while many other works gave the opposite conclusion [15,21]. Even more, similar genetic diversity of non-native populations was also reported [26,27]. It is noted that most of the publications discussed only one invasive plant species to draw a conclusion.

In this work, based on the inconsistent results of the previous studies on the changes of genetic diversity between native and non-native plant populations, a comprehensive meta-analysis was carried out for the re-statistics of those pieces of literature to investigate the general changing trends of IASs’ genetic diversity and explore the correlation of genetic diversity and invasion fate. The result of this work will be helpful to determine whether genetic diversity can be used as an indicator in risk assessment and invasion dynamic prediction of IASs.

## 2. Results

### 2.1. Summary of Database

Firstly, we removed duplicate records and reviews, books, and patents before screening (23 records). Secondly, based on article titles and abstracts, 2908 records not dealing with invasive plants were excluded. Thirdly, articles having parameters such as average value, standard deviation (or standard error), sample size (the number of populations in native or non-native range), and other relevant data that were required in the meta-analysis were selected. As a result, only 26 articles were found to have quantitative analyses of the genetic diversity of invasive plants (measured traits: gene diversity, allelic richness, and the number of alleles or expected heterozygosity) in both native and non-native populations (Figure 1), which met with our criteria and dealt with 19 invasive plants using a total of 27 datasets (Table 1). These plants include five annual herbs, one biennial herb, ten perennial herbs, and three woody or shrubby plants, which are native to Asia, America, or Europe. Among them, *Acacia saligna* and *Phyla canescens* are tropical originated, while the others grow in temperate zone. The different genetic diversity indexes used in those studies included gene diversity (*H*j), allelic richness (*A*_R_), number of alleles (*N*a), and expected heterozygosity (*H*_E_). Fifteen papers used more than one of those indicators, and the highest-ranking indicator was preferably utilized in our work.

### 2.2. Change Trend of Genetic Diversity

A conventional meta-analysis by the inverse-variance weighted method was performed in this study to compare the genetic diversity levels of the IASs between their native and non-native ranges. After testing the heterogeneity among studies derived from the fixed effect model, the weighted mean effect size was found to correlate to the high heterogeneity among each study (*Q* = 155.906, *I*^2^ = 83.323, Tau Squared = 1.008, *p* < 0.001).

Therefore, the random effect model (the mixed effects meta-analysis model) was employed to cope with the high heterogeneity, which implies that we assumed that differences among studies are not only due to sampling error but also due to true random variation, as is the rule for ecological data [38]. The mean effect size obtained with the random effect model is −0.249, while 95% CI ranged from −0.674 to 0.176 with *p* = 0.251 (Figure 2). The results indicated that although a general reduction of IASs’ genetic diversity was found in non-native populations compared to native ones, the difference was not significant.

### 2.3. Reliability of Results

The sensitivity analysis conducted in this study showed no significant change in the mean effect size compared with the previous one, indicating the good stability of our data (Figure 3). The funnel plot, which depicts the distribution of positive and negative datasets used in the meta-analysis, demonstrated a balanced distribution (Figure 4). Furthermore, the results of Egger’s regression test supported this observation (*p* = 0.572 > 0.05). Therefore, these findings suggested the reliability of the meta-analysis conducted in this work, affirming that the results were not influenced by potential biases.

## 3. Discussion

### 3.1. Potential Causes of the Different Changes in Specific IASs’ Genetic Diversity

This study employed a meta-analysis approach, which yielded a negative effect size and a 95% CI overlapping zero based on the 26 selected studies. These results indicate that there is no significant change in IASs’ genetic diversity between native and non-native populations. Although genetic diversity seems not to play a decisive role in determining the fate of invasion, it is important to note that the observed changes in genetic diversity vary among specific species.

The decline of genetic diversity within non-native populations may be due to various causes. Alien species would obtain a small size population after arriving in a new environment; therefore, only a small fraction of the population’s original genetic variation could be carried there, which is called the “founder effect” [39,40]. The founder effect is the extreme form of genetic drift, dominantly influencing the genetic diversity and population structure of the invaders [20], which might lead to the stochastic loss of genetic diversity [41,42].

However, it is noteworthy that the non-native populations of *Bunias orientalis* and *Arctotheca populifolia* did not show a decrease in genetic diversity compared to their native ones, which might be due to the high level of propagule pressure through multiple introduction events [15,17]. Generally, genetic diversity could not only be influenced by population bottleneck, but also by introduction history, mutation, inbreeding, adaptive capacity, and social organization [10,43,44,45].

### 3.2. Decreased Genetic Diversity Does Not Necessarily Affect Invasion Process

Based on the current hypothesis, IASs encounter the founder effect and population bottleneck during their introduction into a non-native range [24,31], resulting in the reduction of genetic diversity, which might limit their viability and evolutionary or adaptive potential to the new environment [22,46]. However, according to our analysis as well as those articles collected from the ISI-Web Science database, it was found that the introduced populations having decreased genetic diversity could still adapt to the new habitats and invaded the non-native areas successfully within 40 years or less [16,27], the underlying mechanism of which still needs further exploration.

According to the hypothesis of ‘pre-adaptation’, when the new habitats are similar to the native ranges, there are few adaptive challenges for the IASs to establish new populations [47,48]. A similar environment is likely to ensure these species maintain their adaptive capacity to survive [49]. Although the genetic diversity of IASs may be reduced after introduction, their fitness to the new environment can be hardly affected. On the other hand, when the environmental conditions of the new ranges are different from the native ones, the loss of genetic diversity could definitely bring adaptive challenges for the introduced populations. In this case, various mechanisms might have been evolved by the IASs to increase their fitness for better adaptation. First of all, diversity loss could have resulted from bottleneck events and/or natural selection, which might promote the rapid evolution of the IASs to increase population fitness [50,51,52]. Secondly, the deleterious alleles causing inbreeding depression might be purged by bottleneck events, which is beneficial to the invasion process [48]. Thirdly, the multiple introductions commonly occurring during invasion could bring about large amounts of genetic variation and novel genetic combinations [53]. The fitness of introduced populations, thus, could be maintained and result in range expansion [54]. It was reported that except for genetic diversity, the plasticity of IASs was another key factor for tolerance enhancement under environmental pressure [32]. The high phenotypic plasticity of IASs could enable the invasive population to move toward a new optimal adaptive range despite their low genetic variation [55,56]. Epigenetic changes are crucial for the plasticity of invasive species [48], which might weaken the negative effects of bottlenecks on adaptability [57]. In addition, mutation is another strategy to enrich the genetic variation and alleviate the adaptive challenge for IASs in a relatively short time period of biological invasion [58,59].

Taken together, the decrease in genetic diversity within non-native populations does not necessarily mean a decrease in invaders’ fitness or much less failure of invasions.

### 3.3. Genetic Diversity Is Not a Useful Indicator in Risk Assessment of IAS

A successful invasion is roughly divided into three stages: introduction, establishment, and expansion [60,61]. Biological invasion can get stuck at any of these stages [61], all of which are our opportunities to terminate the invasive process. Invasiveness is an important feature of IAS, reflecting their capacity of overcoming various barriers at each stage [62]. High genetic diversity was shown to be beneficial for the formation of new genotypes with high invasiveness [63]. But other studies reported genetic diversity decrease was irrelevant to the progress of invasion and establishment [16,27]. Our finding agrees with the latter, in that IASs’ genetic diversity in invaded areas had a non-significant reduction trend, indicating that genetic diversity is not a key factor for successful invasion. As a matter of fact, invasiveness is affected by numerous factors such as introduction history, facilitation, dispersal dynamics (including long-distance dispersal), propagule pressure, phenotypic plasticity, and rapid evolution [64,65]. Therefore, a combination of these determinants instead of only one should be considered when assessing the invasiveness of alien species [66]. In conclusion, the genetic diversity present in the introduced populations might not be a useful indicator in the invasiveness assessment and the prediction of successful colonization.

Prevention, eradication, and control are three successive steps in IAS management proposed by the CBD (Secretariat to the Convention on Biological Diversity 2001). However, once a non-native species is established in a new region, it is extremely difficult to be eradicated or control [67]. Risk assessment and prevention before the outbreak of IAS are more cost-effective than the eradication of an established population [7,68,69]. Risk assessment is a predictive system that quantitatively and/or qualitatively evaluates risks to provide valuable information on the likelihood and fate of a biological invasion [70,71]. However, up to date, the present risk assessment is still not accurate enough for the prediction [71,72]. A population genetic approach has been proposed to elucidate the genetic variation of IAS, for estimating invasion dynamics for risk assessment schemes [16,67,73]. However, the non-significant difference found in this work indicated that the change in population genetic diversity had no significant effect on the success of plant invasion. In fact, a comprehensive risk assessment addressing both past and future trends of spreading, pathway, evolutionary change, and expansion of IAS is needed to predict and prevent future invasion, but genetic diversity is not a necessary index in risk assessment of invasive species [1].

### 3.4. Recommendations for Future Research

The number of comparative research studies focusing on invasive alien plant species has witnessed a steady increase over the past few decades. These studies have primarily aimed to measure various traits exhibited by these plants in both their native and non-native ranges. However, despite this growing body of research, the investigation of genetic diversity in invasive alien plants between their native and non-native populations remains relatively scarce in this study because of the screening criteria. Moreover, the existing studies display a considerable degree of heterogeneity in terms of the study design. As a consequence of this limited number of studies and high heterogeneity, the statistical ability to detect significant differences in the mean effect size from zero is currently constrained. It is essential to address these limitations to ensure robust and reliable findings in this field of research. To achieve this, we propose that future research endeavors should encompass a larger number of invasive alien plant species and incorporate group-specific comparisons. By broadening the scope of the study and including multiple species, it will be possible to draw more comprehensive and robust generalizations regarding the genetic diversity of invasive alien plants. Furthermore, it is worth noting that the mean effect size has been found to correlate with the high heterogeneity observed in each study. This correlation highlights the need for further investigation into the underlying factors contributing to this heterogeneity. By gaining a deeper understanding of the sources of variation among studies, we can refine our research methodologies and improve the comparability and reliability of the findings.

Moreover, comprehending the underlying reasons behind the diminished genetic diversity in prosperous invasive alien plant species is crucial for comprehending their invasive mechanisms and enhancing the effectiveness of weed risk assessment systems. In light of our preliminary findings, we further propose that future research should prioritize the investigation of key genes that regulate phenotypic traits, rather than focusing solely on the genetic variation of invasive alien plants.

## 4. Materials and Methods

### 4.1. Literature Search and Data Collection

The ISI-Web of Science database was used to search the peer-reviewed publications on the genetic diversity of invasive plants from January 2000 to May 2022 with the keywords including (invasive OR alien OR introduced OR invasion) AND (comparison OR comparative OR compare) AND (plant OR plants) AND (varia* OR divers* OR richness OR polymorph*) AND (gene OR phenot*) AND (establ* OR adapt* OR inva*). As a result, a total of 3543 articles were retrieved. Meanwhile, we obtained 14 relevant papers from other sources.

The following 3 criteria were applied to screen the publications: (1) they were original research papers rather than reviews, books, or patents; (2) their subjects were invasive plant species; (3) and the parameters including the average, standard deviation (or standard error), sample size (the number of populations in native or non-native range), and other relevant data required for the meta-analysis could be directly obtained or calculated from articles. Finally, only 26 out of the 3557 articles were found to meet the above criteria. These papers include quantitative analyses of the genetic diversity of IASs (measured as the traits: gene diversity, allelic richness, and the number of alleles or expected heterozygosity) in both native and non-native populations, from which we summarized 27 datasets of 19 invasive alien plants (Table 1) for the following meta-analysis. 

### 4.2. Statistical Analyses

A meta-analysis is a re-statistical tool for generalization from a collection of inconsistent experimental results reported by different researchers on one topic, which has been widely used for resolving discrepancies in genetic studies [74]. This approach could overcome the limitations of individual studies and increase statistical strength and precision [75,76]. After collecting enough relevant articles, the corresponding formula of statistical indicators in the articles could be applied for further analysis to work out the relationships between target variables [77,78]. Each study could be treated as a sample, and each correlation coefficient based on the sample size of each study should be weighted. Commonly, a standard meta-analysis is to estimate the overall effect size of those independent studies based on the same assumptions [79]. 

There are two models for meta-analysis including the fixed effect model and the random effect one. The major assumption of a fixed effect model is that all effect sizes share a common mean value; thus, the variation among data is solely caused by the sampling error. However, this assumption is unrealistic for most biological meta-analyses, especially those involving multiple populations, species, and/or ecosystems [80,81]. The random effect model gives up the strict assumption that all studies are based on samples from the same underlying population, meaning that this model is fitful for different studies which quantified different underlying mean effects, especially when one study yielded an effect different from other works [82]. If the results of the two models are consistent, or the mean effect size of the fixed effect model is associated with low heterogeneity, the fixed effect model will be chosen for meta-analysis. On the contrary, if two results are significantly different, and the mean effect size of the fixed effect model is associated with high heterogeneity, the random effect model needs to be fitted.

We extracted the parameters including the average (x), standard deviation (*Sd*), and sample size (*n*) of the control group (native populations) and the treated group (non-native populations) from each study of the included articles. If only the standard error (*Se*) is provided only, the following equation was used to transform it to *Sd*:$$Sd=Se\sqrt{n}.$$

We chose the standardized mean difference (SMD; often referred to as *d*, Cohen’s *d*, Hedges’ *d* or Hedges’ *g*) that was not biased by small sample sizes as corresponding common effect size statistics to analyze the difference between the mean effect sizes of the two groups [82,83]. Hedges’ *d* is the preferred measure of effect size for traditional meta-analysis because it has lower Type I error rates than other measures [38]. Hedges’ *d* was calculated referred to Zhou’s method [84] as:$$d=\frac{{\overset{-}{x}}^{n}-{\overset{-}{x}}^{i}}{S}J,$$
where *S* is the standard deviation within each group and calculated as follows:$$S=\sqrt{\frac{{Si}^{2}\left(Ni-1\right)+{Sn}^{2}\left(Nn-1\right)}{Ni+Nn-2}},$$
and *J* is a weighting factor based on the number of populations (*n*) in each case, calculated as
$$J=1-\frac{3}{4\left(Nn+Ni-2\right)-1}.$$

*Nn* and *Ni* are the population numbers (*n*) of native and non-native ranges, respectively. The variance of Hedges’ *d* was calculated as follows:$$Vd=\frac{Nn+Ni}{NnNi}+\frac{d^{2}}{2\left(Nn+Ni\right)}.$$

If the 95% confidence interval (CI) overlaps zero, it indicates that the result is not significant. Heterogeneity was measured using *I*^2^ statistics in conjunction with *Q*-statistics. When the *Q* value is significant and *I*^2^ > 75, the effect size of the included studies is meant to be heterogeneous and there are differences among them. The heterogeneity between studies is caused by two sources of variability. One is sampling error, which is always present in a meta-analysis because every single study uses different samples. Another is the indeterminate number of characteristics that vary among the studies, such as those related to the characteristics of the samples, variations in the treatment, the design quality, and so on [85]. The random effect model is commonly used to deal with heterogeneity [86], which has a wider CI and its result tends to be more conservative. 

In addition, the existence of publication bias should be noted and a sensitivity analysis should be carried out. Sensitivity analysis is performed by the ‘one study removed’ function, which is helpful for obtaining a realistic perspective of the potential impact of biases [87]. Every time one study is removed, a new meta-analysis is carried out to see if the mean effect size changes. A funnel plot is a simple and effective graphical technique for exploring potential publication bias [88], which displays effect sizes plotted against the sample size, standard error, conditional variance, or some other measures of the precision of the estimate. An ideally unbiased sample will exhibit a cloud of data with the shape of a funnel that is symmetric around the population effect size [77]. All analyses were performed by the Comprehensive Meta-analysis V2 software.

## 5. Conclusions

A meta-analysis was carried out to explore the trend of genetic diversity changing of IASs and its role in influencing the process of invasion in this study. The results revealed a general reduction in genetic diversity during the invasive process, but the difference was not significant. Our finding indicated that population genetic diversity was not a substantial influence on the success of plant invasion. Consequently, the utility of genetic diversity as a commonly employed indicator in risk assessment and the prediction of invasion dynamics for IASs may be limited.

## Figures and Tables

**Figure 1 plants-12-02690-f001:**
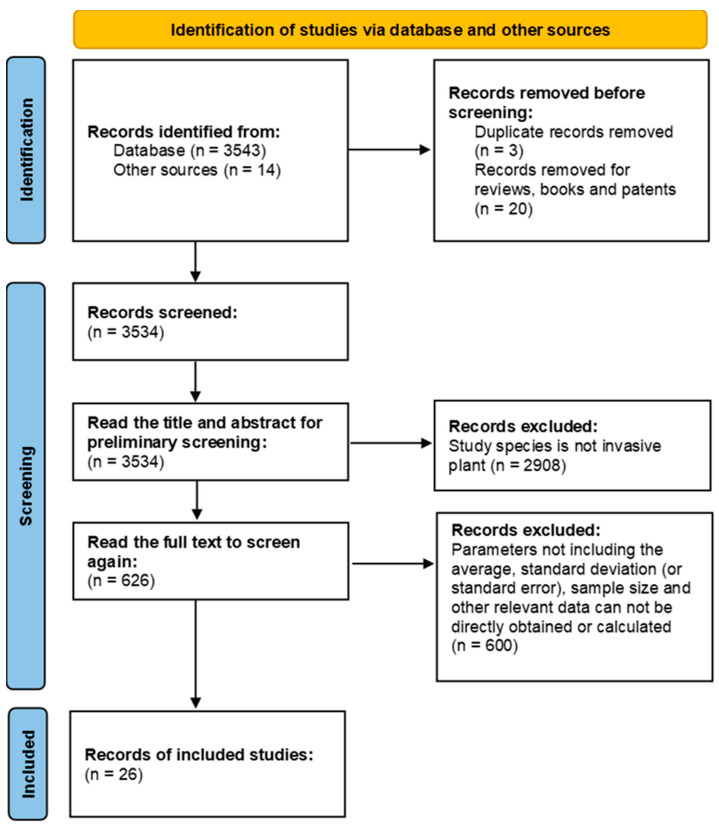
A flowchart summarizing the selection of the literature in meta-analysis.

**Figure 2 plants-12-02690-f002:**
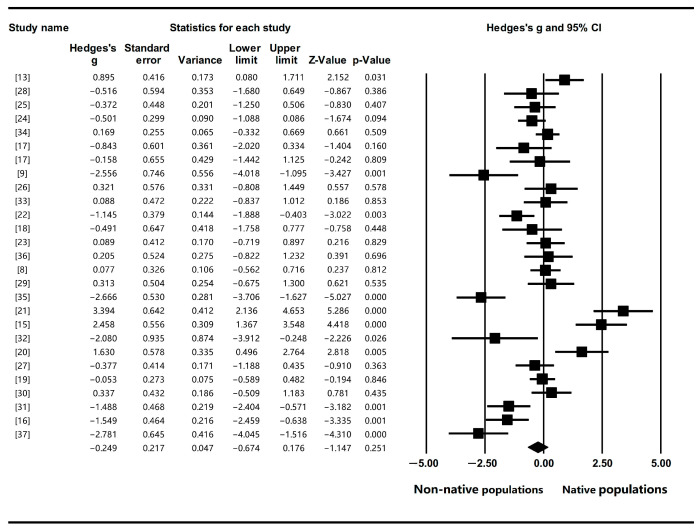
The mean effect size (hedge’s g) of invasive plants’ genetic diversity in their native and non-native range. Note: non-native populations were considered as the treated groups; native populations were considered as the control groups.

**Figure 3 plants-12-02690-f003:**
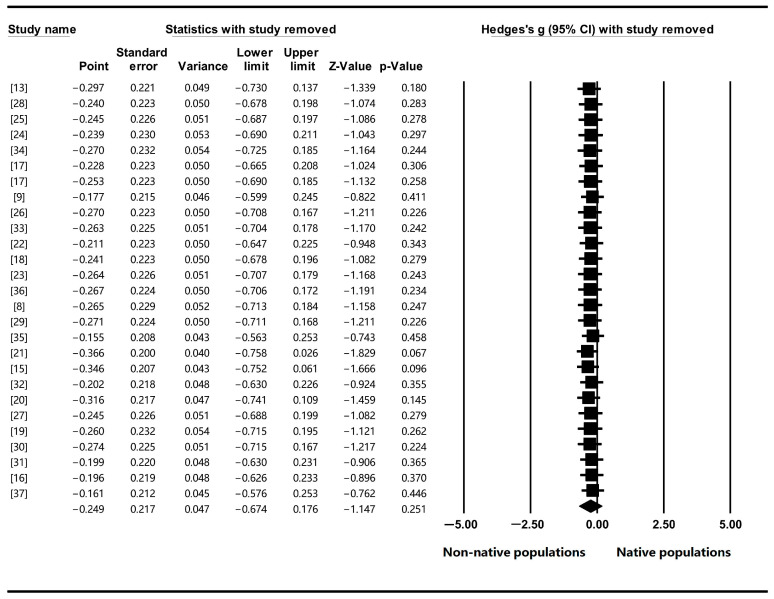
Sensitivity analysis by removing one study.

**Figure 4 plants-12-02690-f004:**
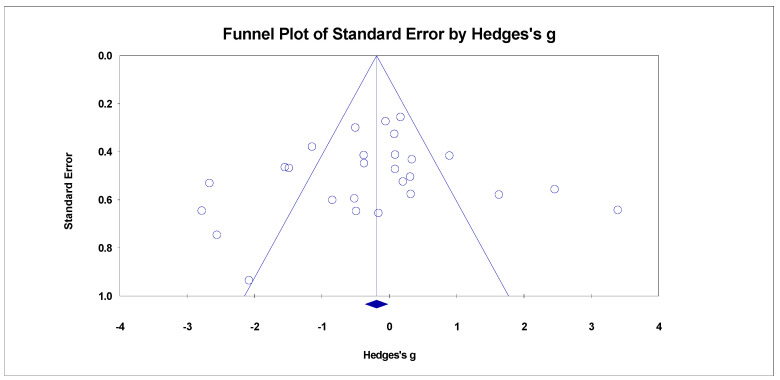
The funnel plot of publication bias of studies on genetic diversity in invasive plants. Note: the dots denote each study; the rhomboid indicates the mean effect size.

**Table 1 plants-12-02690-t001:** Meta-analysis database of invasive plants.

Species	Life Form	Life History	Native Range	Origin	Trait	Refs.
*Acacia saligna*	Woody	Perennial	Australia	Tropics	*N*a	[25]
*Ambrosia artemisiifolia*	Herb	Annual	America	Temperate	*H*j	[28]
*Ambrosia artemisiifolia*	Herb	Annual	America	Temperate	*N*a	[29]
*Ambrosia artemisiifolia*	Herb	Annual	America	Temperate	*N*a	[30]
*Arctotheca populifolia*	Herb	Perennial	Africa	Temperate	*A* _R_	[17]
*Arctotheca populifolia*	Herb	Perennial	Africa	Temperate	*A* _R_	[17]
*Bunias orientalis*	Herb	Perennial	Europe	Temperate	*H*j	[15]
*Centaurea stoebe*	Herb	Perennial	Europe	Temperate	*A* _R_	[8]
*Centaurea stoebe* subsp. *micranthos*	Herb	Perennial	Europe	Temperate	*H* _E_	[13]
*Cirsium arvense*	Herb	Perennial	Eurasia	Temperate	*A* _R_	[24]
*Chromolaena odorata*	Herb	Perennial	America	Temperate	*A* _R_	[9]
*Chromolaena odorata*	Herb	Perennial	America	Temperate	*N*a	[31]
*Erigeron canadensis*	Herb	Annual	America	Temperate	*H* _E_	[19]
*Cotoneaster franchetii*	Woody	Perennial	Asia	Temperate	*H* _E_	[26]
*Helianthus annuus*	Herb	Annual	America	Temperate	*N*a	[20]
*Isatis tinctoria*	Herb	Biennial	Eurasia	Temperate	*H*j	[32]
*Medicago sativa*	Herb	Perennial	Eurasia	Temperate	*H* _E_	[33]
*Oxalis pes-caprae*	Herb	Perennial	Africa	Temperate	*N*a	[23]
*Pinus strobus*	Woody	Perennial	America	Temperate	*H*j	[34]
*Phyla canescens*	Herb	Perennial	America	Tropics	*H* _E_	[22]
*Senecio vulgaris*	Herb	Annual	Europe	Temperate	*H* _E_	[35]
*Sisymbrium austriacum*	Herb	Annual	Europe	Temperate	*N*a	[21]
*Spartina alterniflora*	Herb	Perennial	America	Temperate	*N*a	[18]
*Spartina alterniflora*	Herb	Perennial	America	Temperate	*H*j	[36]
*Spartina alterniflora*	Herb	Perennial	America	Temperate	*H* _E_	[27]
*Spartina alterniflora*	Herb	Perennial	America	Temperate	*A* _R_	[16]
*Spartina alterniflora*	Herb	Perennial	America	Temperate	*A* _R_	[37]

Note: *H*j, gene diversity; A_R_, allelic richness; *N*a, number of alleles; H_E_, expected heterozygosity.

## Data Availability

The data presented in this study are available in Table 1.

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
