# Peer review of "The General Trends of Genetic Diversity Change in Alien Plants’ Invasion"

_plants, 2023, doi:10.3390/plants12142690_

Round 1
Reviewer 1 Report
The study is sound and interesting. This an important topic in the ecology and evolution of invasive species.
The authors have addressed, through meta-analysis, whether genetic diversityof alien plants changes during invasion, relative to native ones. They found in the literature 26 studies that met the criteria for comparison, which belong to 18 plant species. Their statistical results did not revealed differences in genetic diversity between non-native and native populations of species.
The authors conclude “Our finding indicated that population genetic diversity had no significant effect on the success of plant invasion and genetic diversity may not be a useful indicator in risk assessment and invasion dynamic prediction of IASs.” I think they should be more careful about this, given the limited number of studies useful for the analysis, and the dominance of perennial herbs in the sample. I am not saying that the results would change if other plants of different life histories are included but it is worth to assess this.
Finally, what if species with nil genetic variation have vanished in the process of invasion? We do not have a precise idea, yet, how prevalent is failure to invade and the role of genetic variation.
Reviewer 2 Report
The manuscript (ID plants-2491554) deals with the problem of alien plant invasiveness and the role of population genetic diversity in this process. Authors carried out meta-analysis of literature to assess the role of genetic diversity as an indicator in invasion dynamics prediction. The analysis revealed that population genetic diversity may not be a useful indicator in risk assessment and invasion prediction of alien plants. The study is interesting, and the topic is relevant because alien plant invasions cause great economic losses and the reduction of biological diversity. The recommendations related to indicators of accurate prediction of invasions are urgent. In my opinion, the main limitation of this analysis is the rather small number (26) of data (articles) and species (18), included in immediate analysis. For example, Nybom (2004) made compilation of 307 studies, Hamrick and Godt (1996) used data of 735 entries, Pyšek et al. (2009) considered 1218 seed plant species. I understand that every study is different, and that you may have used all the material available at a given time to analyze this issue in a particular aspect. Because of this, I suggest considering (may be in discussion and abstract) your results as preliminary, pilot study, and make conclusions based on this limitation. I would recommend noting that more experimental studies with a larger number of species should be analyzed to draw more robust generalizations.
Specific comments:
The bibliography needs to be corrected because the titles of the journals are missing.
Table 1. Bunias orientalis is a perennial plant (see Dietz et al. 1999, Oecologia).
Line 166. “…species [47],which might...“
Reviewer 3 Report
Overall, I found the manuscript to be well-written.
Why are results given before Materials and Methods?
line 127: ...non-native populations may be due to various causes.
The normal order of information in the References is:
Author(s), title, journal, year, vol. pages
Abbreviations for journals are too short and unrecognizable for most. e.g. in citation 1: B.R. ?? Biol. Rev.
Spell out or abbreviate journals so reader knows where information is published without having to use Google Scholar or some other method.
